# Cardiac risk stratification in cancer patients: A longitudinal patient–patient network analysis

Yuan Hou[1☉], Yadi Zhou[1☉], Muzna Hussain[2,3☉], G. Thomas Budd[4], Wai Hong Wilson Tang[1,2,5], James Abraham[4], Bo Xu[2], Chirag Shah[6], Rohit Moudgil[2], Zoran Popovic[2], Chris Watson[3], Leslie Cho[2], Mina Chung[2,5], Mohamed Kanj[2], Samir Kapadia[2], Brian Griffin[2], Lars Svensson[7], Patrick Collier[2,5]*, Feixiong Cheng[1,5,8]*

**1** Genomic Medicine Institute, Lerner Research Institute, Cleveland Clinic, Cleveland, Ohio, United States of America, **2** Robert and Suzanne Tomsich Department of Cardiovascular Medicine, Sydell and Arnold Miller Family Heart and Vascular Institute, Cleveland Clinic, Cleveland, Ohio, United States of America, **3** School of Medicine, Dentistry and Biomedical Sciences, Wellcome-Wolfson Institute for Experimental Medicine, Queen's University, Belfast, United Kingdom, **4** Department of Hematology/Medical Oncology, Cleveland Clinic, Cleveland, Ohio, United States of America, **5** Department of Molecular Medicine, Cleveland Clinic Lerner College of Medicine, Case Western Reserve University, Cleveland, Ohio, United States of America, **6** Department of Radiation Oncology, Cleveland Clinic, Cleveland, Ohio, United States of America, **7** Department of Cardiovascular Surgery, Cleveland Clinic, Cleveland, Ohio, United States of America, **8** Case Comprehensive Cancer Center, Case Western Reserve University School of Medicine, Cleveland, Ohio, United States of America

☉ These authors contributed equally to this work.
* colliep@ccf.org (PC); chengf@ccf.org (FC)

**Data Availability Statement:** The codes written for and data used in this study are available from website: https://github.com/ChengF-Lab/psnCVD.

## Abstract

### Background

Cardiovascular disease is a leading cause of death in general population and the second leading cause of mortality and morbidity in cancer survivors after recurrent malignancy in the United States. The growing awareness of cancer therapy–related cardiac dysfunction (CTRCD) has led to an emerging field of cardio-oncology; yet, there is limited knowledge on how to predict which patients will experience adverse cardiac outcomes. We aimed to perform unbiased cardiac risk stratification for cancer patients using our large-scale, institutional electronic medical records.

### Methods and findings

We built a large longitudinal (up to 22 years' follow-up from March 1997 to January 2019) cardio-oncology cohort having 4,632 cancer patients in Cleveland Clinic with 5 diagnosed cardiac outcomes: atrial fibrillation, coronary artery disease, heart failure, myocardial infarction, and stroke. The entire population includes 84% white Americans and 11% black Americans, and 59% females versus 41% males, with median age of 63 (interquartile range [IQR]: 54 to 71) years old.

We utilized a topology-based K-means clustering approach for unbiased patient–patient network analyses of data from general demographics, echocardiogram (over 25,000), lab testing, and cardiac factors (cardiac). We performed hazard ratio (HR) and Kaplan–Meier analyses to identify clinically actionable variables. All confounding factors were adjusted by

**Funding:** This work was supported by the National Heart, Lung, and Blood Institute of the National Institutes of Health (NIH) under Award Number K99HL138272 and R00HL138272 to F.C. This work was supported in part by the National Institute of Aging (R01AG066707 and 3R01AG066707-01S1) and by the VeloSano Pilot Program (Cleveland Clinic Taussig Cancer Institute) to F.C. The funders had no role in study design, data collection and analysis, decision to publish, or preparation of the manuscript.

**Competing interests:** The authors have declared that no competing interests exist.

**Abbreviations:** AF, atrial fibrillation; AMI, adjusted mutual information; ARI, adjusted rand index; ASE, American Society of Echocardiography; AUROC, area under the receiver operating characteristic curve; BH, Benjamini and Hochberg; BMI, body mass index; BSA, body surface area; CAD, coronary artery disease; CI, confidence interval; CTRCD, cancer therapy–related cardiac dysfunction; CVD, cardiovascular disease; EDV, end-diastolic volume; EMR, electronic medical records; ESV, end-systolic volume; HF, heart failure; HR, hazard ratio; ICD, International Classification of Diseases; IQR, interquartile range; KM, Kaplan–Meier; KS, Kolmogorov–Smirnov; LVEF, left ventricular ejection fraction; MI, myocardial infarction; NT-proBNP, NT-proB-type Natriuretic Peptide; PCC, Pearson correlation coefficient; psnCVD, patient–patient similarity network-based risk assessment of CVD; REDCap, research electronic data capture; SSE, sum of squared error.

Cox regression models. We performed random-split and time-split training-test validation for our model.

We identified 4 clinically relevant subgroups that are significantly correlated with incidence of cardiac outcomes and mortality. Among the 4 subgroups, subgroup I ($n = 625$) has the highest risk of de novo CTRCD (28%) with an HR of 3.05 (95% confidence interval (CI) 2.51 to 3.72). Patients in subgroup IV ($n = 1,250$) had the worst survival probability (HR 4.32, 95% CI 3.82 to 4.88). From longitudinal patient–patient network analyses, the patients in subgroup I had a higher percentage of de novo CTRCD and a worse mortality within 5 years after the initiation of cancer therapies compared to long-time exposure (6 to 20 years). Using clinical variable network analyses, we identified that serum levels of NT-proB-type Natriuretic Peptide (NT-proBNP) and Troponin T are significantly correlated with patient's mortality (NT-proBNP > 900 pg/mL versus NT-proBNP = 0 to 125 pg/mL, HR = 2.95, 95% CI 2.28 to 3.82, $p < 0.001$; Troponin T > 0.05 μg/L versus Troponin T $\leq$ 0.01 μg/L, HR = 2.08, 95% CI 1.83 to 2.34, $p < 0.001$). Study limitations include lack of independent cardio-oncology cohorts from different healthcare systems to evaluate the generalizability of the models. Meanwhile, the confounding factors, such as multiple medication usages, may influence the findings.

## Conclusions

In this study, we demonstrated that the patient–patient network clustering methodology is clinically intuitive, and it allows more rapid identification of cancer survivors that are at greater risk of cardiac dysfunction. We believed that this study holds great promise for identifying novel cardiac risk subgroups and clinically actionable variables for the development of precision cardio-oncology.

## Author summary

### Why was this study done?

- An increasing number of oncology patients are facing cancer therapy–related cardiac dysfunction (CTRCD) risk, leading to the emerging field of cardio-oncology (also known as onco-cardiology); however, there are limited clinical guidelines in terms of how to prevent and treat for the new cardiotoxicity among cancer survivors.

- Development of novel clinical tools would offer unique opportunities for precision cardio-oncology by utilizing the large-scale, longitudinal patient data from healthcare systems.

### What did the researchers do and find?

- We developed a longitudinal patient–patient network clustering methodology for cardiac risk stratification in cancer patients during anticancer therapies.

- We identified 4 clinically relevant subgroups that are statistically significantly correlated with incidence of cardiac outcomes and all-cause mortality.

- Using longitudinal patient–patient network analyses (over 20 years' follow-up), we showed crucial roles of early cardiovascular care in improving quality of life of cancer survivors and reducing incidence of CTRCD.

- We identified multiple clinically relevant predictors (including Troponin T and NT-proB-type Natriuretic Peptide (NT-proBNP)) that are significantly correlated with incidence of cardiac outcomes and patients' mortality, which offers actionable biomarkers for rapid risk assessment of cardiac dysfunction during cardio-oncology clinical practices.

### What do these findings mean?

- Our findings suggest that an unbiased, systems-based network analysis of large-scale, longitudinal patient data is more interpretable, visualizing the decision boundary to cardiac risk stratification for patients before, during, and after cancer treatment.

- Troponin T and NT-proBNP offer clinically actionable biomarkers for cardiac risk stratification in cardio-oncology clinical practices. Extended independent cohort validations are needed before the predictors are introduced to clinical implementation.

## Introduction

The improvement in early detection and effective oncological treatment has led to an increased number of cancer survivors in the United States [1]. This number is estimated to increase from 16.9 million in 2019 to 22.1 million by 2030 [2]. However, improved survival from cancer leads to greater risk from other life-threatening conditions and, in particular, cardiovascular disease (CVD), which is the second leading cause of mortality and morbidity in cancer survivors [1,3]. The increased risk of CVD in cancer survivors is in part associated with cancer therapy–related cardiac dysfunction (CTRCD) [4], including radiotherapy [5], cytotoxic chemotherapy [6], targeted therapies [7–9], and immunotherapy [10–12]. For example, doxorubicin is the first-line anticancer drug for multiple malignancies; however, doxorubicin has adverse short- and long-term cardiovascular effects including heart failure [13], cardiomyopathy [14], and left ventricular dysfunction [15,16].

The growing awareness of CTRCD has led to the emerging field of cardio-oncology [17]. However, there are limited guidelines in terms of how to assess for, prevent, and treat CTRCD in cancer survivors due to lack of predictive and prognostic assays. Echocardiogram is the most utilized clinical test to assess for CTRCD. The American Society of Echocardiography (ASE) have defined cardiac dysfunction as a reduction in left ventricular ejection fraction (LVEF) >10% below the lower limit of normal [18]. However, traditional echocardiogram approaches alone have limitations including high false positive rates [19]. Additionally, it is already late for intervention when decreased LVEF is recognized, as only 42% patients have partial or full recovery in left ventricular function [20]. Next-generation machine learning technologies can harness the power of large-scale clinical data and offer new possibilities to

predict which patients are at risk and allow for early intervention to prevent risk of CVD. Previously, Samad and colleagues built supervised machine learning models from echocardiogram data and clinical data to predict patient survival [21]. However, traditional "black box" machine learning methods and statistical risk models have various limitations, reducing their ability to predict clinical outcomes in new scenarios from heterogeneous patients [22–24].

Recent advances in artificial intelligence [25] and network science technologies [26–29] offer valuable and increasingly useful network tools for deep phenotyping of patient heterogeneities as seen in patients who developed stroke [30], pulmonary vascular disease [31], as well as those seen in cardio-oncology [10,32–34]. In this study, we utilized a clinically actionable network-based methodology (called patient–patient similarity network-based risk assessment of CVD or psnCVD) for unbiased cardiac risk stratification for cancer patients with CTRCD using large-scale, longitudinal, heterogeneous patient data, including demographics, echocardiogram, laboratory testing, and cardiac factors. With the aid of psnCVD, patients of unknown status can be classified based on their similarity to patients with known status, offering precision medicine approaches to identify patients that are highly sensitive to CTRCD (and allowing more rapid identification of patients that are at greater risk of CTRCD). Compared to traditional supervised risk methods, we hypothesized that our unsupervised psnCVD can leverage heterogeneous patient data and generate interpretable models to visualize the decision boundary in cardiac risk stratification of cancer patients with CTRCD.

## Methods

### Study population and clinical variables

All adult patients with cancer referred to the cardio-oncology service at the Cleveland Clinic from March 1997 up to January 2019. Our retrospective study has not prespecified analysis plan. However, the patient pool in this study represents oncology patients seen by oncology specialists at our institution undergoing cancer treatments and referred for cardiology evaluation/testing based upon cardiac risk factor profile or cardiac comorbidity. Once patients were identified, patient information was collected. This study was reviewed and approved by the Institutional Review Board. In addition, this study is reported as per the STARD 2015 reporting guideline for diagnostic accuracy studies (**S1 Checklist**).

Comprehensive clinical information was collected using the institutional electronic medical records (EMR) database by International Classification of Diseases (ICD 9/10) codes after cancer diagnosis. This cohort of patients is seen at Cleveland Clinic and regularly followed up. Although a minority of cases moved to another institution, the EMR at Cleveland Clinic is part of the Care Everywhere Network, which is used in 373 institutions across 48 states in the US. This allowed us to collect the details of visits from any such institution and therefore analyze relevant outcomes for these patients. For each patient, 112 clinical variables commonly collected during cardio-oncology clinical practices were used in this study (**S1 Table**): (a) 43 general demographics; (b) 24 lab testing variables; (c) 7 cardiac variables; and (d) 38 echocardiogram variables. Echocardiogram clinical variables were generated from a total of 23,451 sequential echocardiograms. Detailed clinical characteristics of the entire cohort used are provided in **Table 1**.

### Outcomes

All-cause mortality with up to 20 years' follow-up data (1997 to 2019, median with interquartile range (IQR) were 5.02 [2.39 to 8.01]) was used as the primary outcome. Cardiac outcomes defined by ICD 9/10 codes were manually checked through looking at patient charts on Epic for accuracy, including atrial fibrillation (AF), coronary artery disease (CAD), heart failure

**Table 1. Baseline characteristics and clinical outcomes.**

| Baseline characteristics of the entire cohort ($n$ = 4,632) | | |
|---|---|---|
| Age (year) | | |
| Median (IQR) | 63 (54–71) | |
| Race | | |
| White Americans | 3,910 | 84% |
| Black Americans | 516 | 11% |
| Other and unknown | 206 | 5% |
| Sex | | |
| Female | 2,739 | 59% |
| Male | 1,893 | 41% |
| BMI (kg/m$^2$) | | |
| Median (IQR) | 27 (23–32) | |
| ≥30 | 1,610 | 35% |
| <25 | 1,645 | 36% |
| 25–29.9 | 1,377 | 30% |
| Tobacco | 2,317 | 50% |
| Family history | 1,654 | 36% |
| Comorbidities | | |
| Hypertension | 2,622 | 57% |
| Hyperlipidemia | 2,010 | 43% |
| Diabetes | 1,039 | 22% |
| Malignancy | | |
| Hematologic cancer | 1,822 | 39% |
| Solid tumor cancer | 2,810 | 61% |
| Clinical endpoints | | |
| Mortality (all cause) | 1,799 | 39% |
| Mortality (in hospital) | 486 | 10% |
| Cardiac events | 1,670 | 36% |
| Pre-existing | 784 | 17% |
| De novo CTRCD | 886 | 19% |

The cohort has 4,632 patients in total. Data for continuous variables were presented as median (IQR), and data for categorical variables were presented as number of percentages, n (%). Cardiac events: 5 hospital diagnosed outcomes by ICD 9/10 codes, including AF, CAD, HF, MI, and stroke. De novo CTRCD: The patient has at least one type of cardiac events diagnosed after cancer therapy.

AF, atrial fibrillation; BMI, body mass index; CAD, coronary artery disease; CTRCD, cancer therapy–related cardiac dysfunction; HF, heart failure; IQR, interquartile range; MI, myocardial infarction.

(HF), myocardial infarction (MI), and stroke. According to the diagnosis date of these 5 cardiac outcomes, we identified the cardiac events diagnosed before cancer therapy as preexisting cardiac events, and those after cancer therapy as de novo CTRCD. All diagnoses defined by ICD 9/10 codes were further confirmed by manual review of all medical records.

## Preprocessing and imputation of clinical variables

Since our echocardiogram and partial general demographics data were longitudinal, for each variable, we extracted several features: maximum of all follow-ups, minimum of all follow-ups, slope of the variable versus time of all follow-ups, maximum increase within 3 months, and

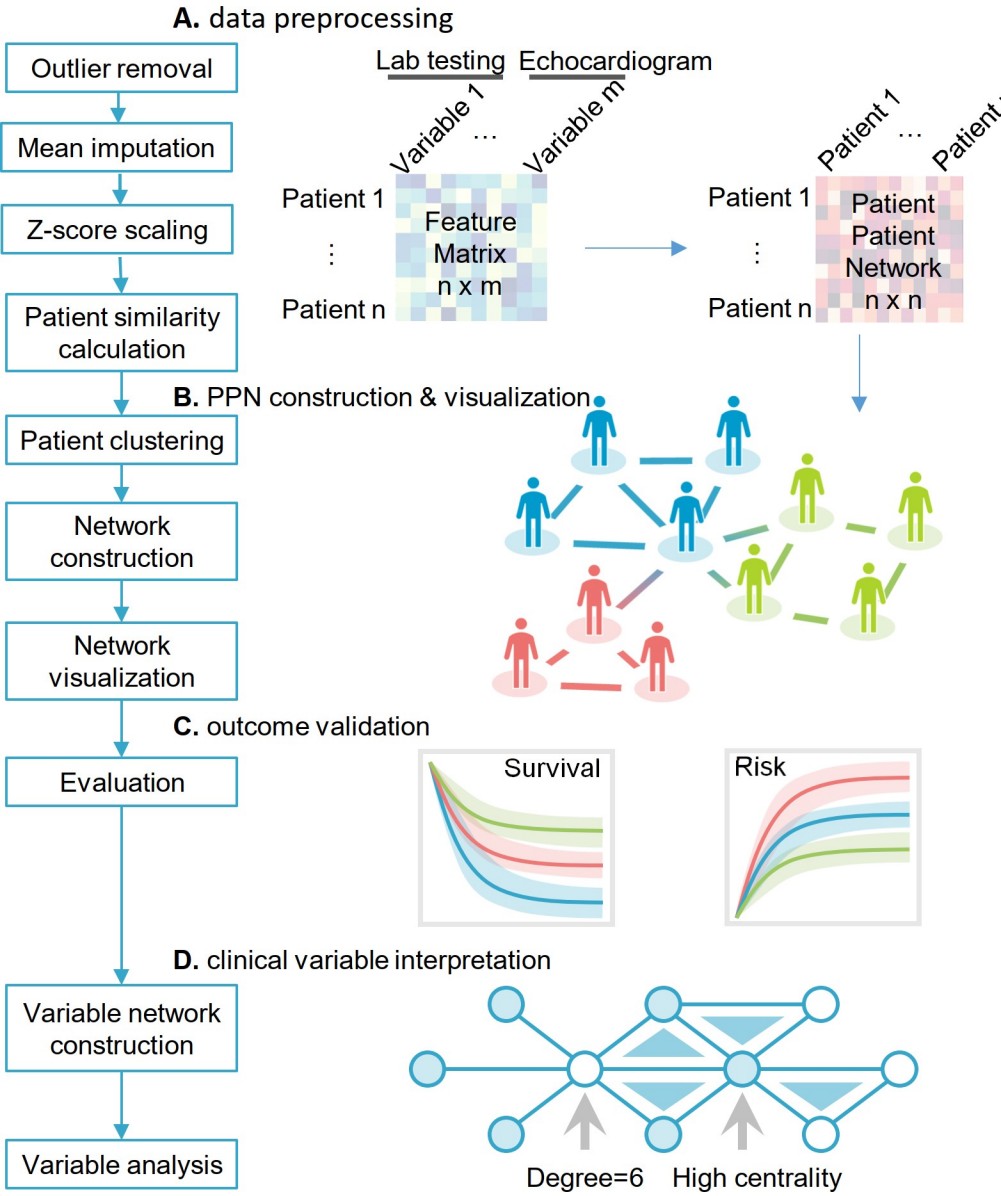

**Fig 1. Overall study design.** The overall study design included 4 steps: (A) data preprocessing; (B) PPN construction and visualization; (C) clinical validation using cardiac outcomes and survival analysis; and (D) clinical variable interpretation. The data preprocessing includes outlier removal, feature scaling by z-score method, and missing data imputation. With the preprocessed patient-clinical variable matrix, we used cosine measure as the similarity metrics for generating a patient–patient similarity network. Then, we performed K-means clustering to layout patients to different subgroups based on the cosine measure (see Methods). Patients with similar clinical characteristics are grouped in the same cluster and are visualized through a specific subgroup to form the final PPN. After the patient network construction and visualization, we used 2 clinical outcomes, mortality and CTRCD to evaluate performance of network-based clustering. Finally, we performed the clinical variable network analysis to enhance clinical interpretation of each risk subgroups with CTRCD. CTRCD, cancer therapy–related cardiac dysfunction; PPN, patient–patient network.

maximum decrease within 3 months. In total, we obtained 112 variables (including the derived ones). A detailed description for all the variables can be found in the supplemental methods section (**S1 Table**). In this study, 4,632 patients were kept for downstream analysis. Missing values were imputed using the mean method, followed by z-score scaling (**Fig 1**).

## Construction of a patient–patient similarity matrix

For the construction of the patient–patient network, we computed the cosine similarity for all pairs of patients (**Fig 1**). The cosine similarity of patient $A$ and $B$ was calculated as:

$$\text{cosine}_{AB} = \frac{\sum_{i=1}^{n} A_i B_i}{\sqrt{\sum_{i=1}^{n} A_i^{2}} \sqrt{\sum_{i=1}^{n} B_i^{2}}} \tag{1}$$

where $n = 112$, and $A_i$ and $B_i$ indicate the $i^{th}$ variable of patient $A$ and $B$, respectively. A cosine cutoff was used to determine if 2 patients should be connected in the network for visualization.

## Network clustering

To identify patient subgroups, we clustered the 4,632 patients using their cosine similarity network profiles by K-means clustering analysis (**Fig 1**). We first tried to use the elbow method [35] to determine the number of clusters. We tested the range of 3 to 20 of the sum of squared error (SSE):

$$\text{SSE} = \sum_{i=1}^{n}(X_i - \bar{X})^2 \tag{2}$$

where $X_i$ indicates each patient, and $\bar{X}$ is the average of the patients within the cluster. However, SSE was decreasing smoothly as the number of clusters increase. Therefore, we performed the survival analysis and cardiovascular outcome analyses for different number of clusters to identify the best K value. In this study, we chose the best cluster number (K = 4) using subject matter expertise based on a combination of factors (log-rank $p < 0.05$; **S1 Fig and S2 Table**): (i) significantly distinguishable survival rate and cardiovascular outcome by Kaplan–Meier (KM) estimator with log-rank test; and (ii) the highest number of clusters to identify more new patient subgroups. For each cluster, we computed the ratio of patients with CVD and the $p$-value using a $\chi^2$ test.

Considering that the K-means clustering has a stochastic component, which may result in different clusters being produced from the same input data, we computed the adjusted rand index (ARI) and adjusted mutual information (AMI) to validate the clustering stability [36,37]. For both metrics, a value of 1 indicates perfect agreement, while randomly assigned clusters have scores around 0. Following the workflow (**S2A Fig**), we performed 100 K-means clustering experiments using different random initial states. Among the 100 random experiments, 99 showed high ARI and AMI scores for the clusters, indicating robustness of the clustering results (**S2B Fig**).

## Network visualization

To better visualize the patient–patient networks, we computed the network density at different cutoff values and selected the cutoff that resulted in the lowest network density [38,39]. Network density is defined as the ratio of the number of actual links and the number of all possible links from all the patients. The number of all possible links is calculated as n × (n − 1) / 2, where n is the number of patients in the network. Using this method, we tested the cutoffs in an increment of 0.05 and identified that the lowest network density (0.24%; **S3 Fig**) was achieved when the cutoff was 0.65. Finally, all patient pairs with cosine similarity >0.62 were considered connected in the network to retain more patients for the network visualization and obtain a lower network density. In addition to cosine similarity, we also tested Pearson correlation coefficient (PCC), but this latter measure was not able to yield more distinguishable

clusters (**S4 Fig**). The density minimization procedure was used to optimize a network layout, which does not have a direct impact to improve performance of patient network clustering. The patient network with each cluster indicated by a color was visualized using Cytoscape v3.7.1 [40].

## Variable network construction

In order to understand the differences among the patient subgroups in terms of the clinical variables, we constructed a clinical variable network for each patient subgroup. For each cluster, PCC values of all pairs of noncategorical variables using their distribution in the patients within a specific subgroup were calculated. For the derived echocardiogram variables, the maximum absolute PCC was used to represent the correlations between these variables and other non-echocardiogram variables. However, there were a limited number of variables; the network density–based PCC cutoff selection strategy resulted in very sparse networks with too few variables present in the network. Therefore, we adopted a top K percent strategy that uses the K% connections with the highest PCC for the construction of the network. To determine which K to use, we test the following percentages: 5%, 10%, 15%, and 20% (**S5 Fig**). For example, using top 5%, all variable pairs with |PCC| greater than the absolute PCC at the top 5% were connected. Too few clinical variables were still present in the network when 5% and 10% were used. When 20% was used, we found an increasing number of correlations with nonsignificant $p$-values ($p > 0.05$). Therefore, 15% was used for the final clinical variable network analysis. At this cutoff, the highest $p$-value among all the correlations in all clusters was 0.008.

## Network analysis

We utilized the Python 3.7 package NetworkX [41] to investigate the properties of the clinical variable networks and used 2 approaches for evaluation. For clinical variable evaluation, we used node degrees and betweenness centrality to rank the variables in the networks. We then checked whether some clinical variables (nodes) were important to the network. We used a complete linkage hierarchical clustering algorithm to cluster the variables across four subgroups.

## Statistical analysis

The KM method was used to estimate probabilities of overall survival of the 4 subgroups. The survival rate was calculated from the cancer start date to death (all-cause), and log-rank test was used for comparison among different subgroups with Benjamini and Hochberg (BH) adjustment [42]. All the survival analyses were performed using the Survival and Survminer packages in R v3.6.0 (https://www.r-project.org). Statistical tests for assessing cardiac outcome enrichment across different subgroups through χ2 were performed by SciPy v1.2.1 (https://docs.scipy.org/doc/scipy/reference/index.html). The Kolmogorov–Smirnov (KS) test was used to assess continuous variable comparisons, and one-way ANOVA was used to compare the difference of clinical variables among 4 subgroups. $p < 0.05$ was considered statistically significant. All confounding factors (including age, sex, tumor types, tumor stages, disease comorbidities [e.g., hypertension and diabetes], and medications) were adjusted by Cox regression models.

## Results

### Cohort description

The study cohort contains 4,632 cancer patients with at least 2 follow-up visits from March 1997 to January 2019 at the Cleveland Clinic (**Table 1**). In addition to the clinical data from

each patient, data from a total of 23,451 echocardiograms were collected (including baseline and longitudinal follow-up studies). The overall population are 59% females and 41% males, among which 39% were diagnosed with a hematologic cancer, and 61% with solid tumors at their initial cancer diagnosis (**Table 1**). The median age is 63 (IQR: 54 to 71) years old for the overall population. Median body mass index (BMI) is 27 kg/m$^2$ (IQR: 23 to 32 kg/m$^2$), and there were 1,610 (35%) patients with BMI $\geq$30 kg/m$^2$ (in obese range). Overall, 1,799 (39%) patients died during the study period, and 486 (10%) patients died in hospital.

In this study, we used 5 types of cardiovascular events defined by ICD 9/10 codes and manually checked by looking at patient charts on Epic for accuracy, including AF, CAD, HF, MI, and stroke. In total, 1,670 (36%) of patients have at least one type of diagnosed cardiac event. Specifically, 784 (17%) patients had preexisting cardiac events before cancer therapy, while 886 (19%) patients developed de novo CTRCD. The de novo CTRCD is defined as diagnosed cardiovascular events (AF, CAD, HF, MI, or stroke) after cancer therapy. This number is consistent to the previous research in breast cancer populations, in which 18% of patients were resulted from cardiac dysfunction receiving doxorubicin and trastuzumab [43].

## Network-based discovery of novel cardiac risk subgroups

Using the framework of psnCVD (**Fig 1**), we identified 4 subgroups (clusters; **Fig 2**) that had the most distinct survival rate among the overall cohort. Among 4 subgroups, orange subgroup (C1, $n = 625$; **S3 Table**) and green subgroup (C3, $n = 949$; **S4 Table**) were most significantly enriched with CTRCD: 51% (95% confidence interval [CI] 47% to 54%, $p < 0.001$, $\chi^2$ test) of patients in orange subgroup (28% de novo CTRCD) and 46% (95% CI 43% to 49%, $p < 0.001$, $\chi^2$ test) of patients in green subgroups (24% de novo CTRCD), respectively. Blue subgroup (C2) was the largest subgroup in the patient–patient network (1,808 patients; **S5 Table**), while the CTRCD percentage was only 24%, indicating the lowest CTRCD risk subgroup. In purple subgroup (C4, $n = 1,250$; **S6 Table**), 39% of cancer patients had CTRCD.

To better evaluate the clinical relevance of patient–patient networks, we performed KM analysis to estimate cumulative hazard of de novo CTRCD and survival rate across 4 network-predicted subgroups (**Fig 2B** and **2C**). A higher cumulative hazard of de novo CTRCD indicates a higher incidence of CTRCD after cancer therapy initiation. The cumulative hazard of de novo CTRCD gradually increases from blue, purple, green, to orange subgroups (log-rank, $p < 0.001$; **Fig 2B**), and the hazard ratios (HRs) show the same trend as well (**Fig 2D**). Among 4 subgroups, orange subgroup has the highest risk of de novo CTRCD (**Fig 2D**) with an HR of 3.05 (95% CI 2.51 to 3.72, $p < 0.001$). Conversely, blue subgroup has the lowest CTRCD risk (**Fig 2B**).

To further test the performance of the risk stratification on each of cardiovascular events, we computed the cumulative hazard and percentage of HF, AF, CAD, MI, and stroke across 4 subgroups (**S6 Fig**). To be specific, the orange subgroup (C1) has the highest cumulative hazard of de novo HF and AF (log-rank, $p < 0.001$; **S6A Fig**), while the blue subgroup (C2, lowest CTRCD and mortality rate subgroup) has lowest cumulative hazard of de novo HF, AF, CAD, and MI (log-rank, $p < 0.001$; **S6A Fig**). Yet, the cumulative hazard of de novo stroke is slightly separated across 4 subgroups (log-rank, $p = 0.055$). We found that with the increased cumulative hazard of HF, the percentage of HF from blue, pink, green, to orange were elevated (**S6B and S6C Fig**). To be specific, 19.5% of preexisting and de novo HF patients were in orange subgroup (C1, highest CTRCD subgroup; **S6B Fig**), which is significantly higher than 13.7% of green subgroup (orange 95% CI 16.4% to 22.6% versus green 95% CI 11.5% to 15.9%, $p = 0.011$, $\chi^2$ test). We also found that the cumulative hazard of de novo CAD was not significant between orange and green subgroup (**S6A Fig**). However, the green subgroup had highest percentage of CAD (16.3%), and the percentage of preexisting CAD was 10.6% (**S6B Fig**).

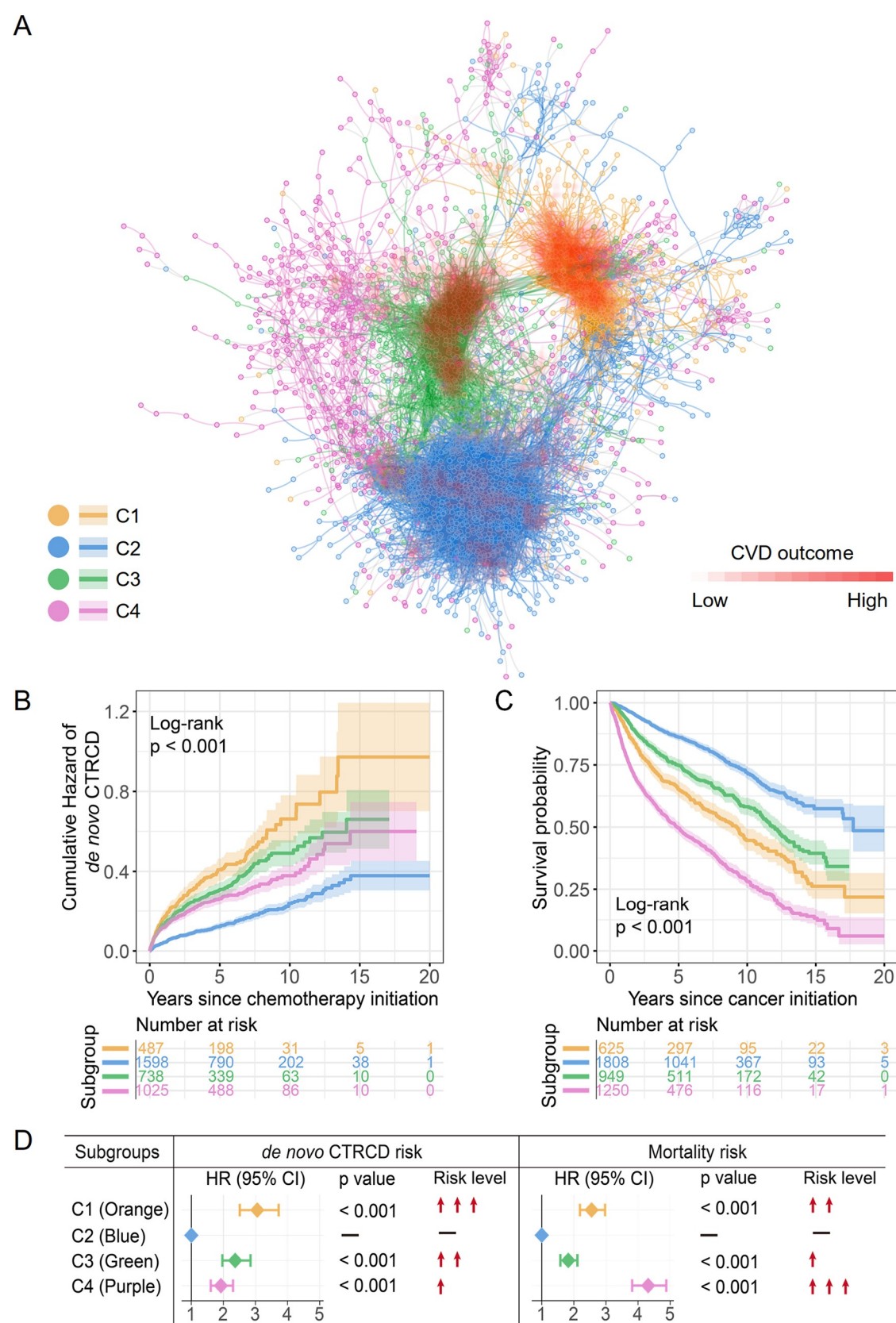

**Fig 2. A discovered patient–patient similarity network.** (**A**) A patient–patient network colorized by 4 clusters (subgroups). In total, 3,131 patients were shown. A total of 15,698 edges with cosine >0.62 were illustrated. The cosine cutoff was selected based upon the network density (**S2 Fig**). A gradient of red color was used to highlight CTRCD outcomes among different patient subgroups, whereby dense red saturation means more enriched outcomes of CTRCD. The network was visualized by Cytoscape v 3.7.1. (**B**) Cumulative hazard of de novo CTRCD (the patient has at least one type of cardiac event diagnosed after cancer therapy) and (**C**) KM curves to estimate all survival probability across 4 subgroups are shown. The log-rank test with the BH adjustment was used for comparing the cumulative hazard of de novo CTRCD and survival rate among 4 subgroups. The shadow represents 95% CI. (**D**) The effects of 4 subgroups with risk of de novo CTRCD and all-cause mortality were estimated with HRs (and 95% CI), and the Wald $\chi^2$ test was used to evaluate the subgroups with statistically significant coefficients. Orange subgroup: C1, intermediate survival and the highest de novo CTRCD risk; blue subgroup: best survival and the lowest de novo CTRCD risk; green subgroup: intermediate survival and intermediate de novo CTRCD risk; purple subgroup: the worst survival and intermediate de novo CTRCD risk. BH, Benjamini and Hochberg; CI, confidence interval; CTRCD, cancer therapy–related cardiac dysfunction; CVD, cardiovascular disease; HR, hazard ratio; KM, Kaplan–Meier.

We next turned to analyze overall survival rate across 4 subgroups. With the de novo CTRCD risk increasing from blue, green, to orange subgroups, the survival probability dropped significantly (log-rank, $p < 0.001$; **Fig 2C**). Specifically, the patients in blue subgroup have the lowest risk of de novo CTRCD and the best survival. Patients in purple subgroup had the second lowest risk of de novo CTRCD (log-rank, $p < 0.001$; **Fig 2B** and **2D**) but the worst survival probability (HR 4.32, 95% CI 3.82 to 4.88; **Fig 2D**). Thus, patients within purple subgroup represented a relatively intermediate CTRCD risk but the worst mortality subgroup.

Among 4 network-predicted subgroups, we found that patients within purple subgroup are heterogeneously distributed across other subgroups (**Fig 2A**). Patients within purple subgroup had a moderate risk of de novo CTRCD (**Fig 2B**), while it was enriched by the worst mortality rate (**Fig 2C**). One possible explanation is that tumor types or tumor stages may influence the mortality. We therefore estimated the HRs of mortality across different tumor types and tumor stages. Cox regression analysis showed that the increased mortality in the purple subgroup was significantly associated with the late tumor stages (HR = 2.07, 95% CI 1.50 to 2.85, $p < 0.001$; **S7 Fig**). However, the different tumor types and tumor stages do not influence the total performance of our network method. The survival and cumulative hazard of de novo CTRCD showed the same results with or without the features of tumor types, tumor stages, and treatment types (**Fig 2B and 2C**, **S8 Fig**).

In addition to K-means clustering on patient–patient similarity networks, we tested performance of K-means clustering using the raw clinical variables for all patients. We found that the 2 cardiac-risk subgroups (the orange subgroup and green subgroup; **S9 Fig**) identified from K-means clustering using the raw clinical variables are not significantly associated with survival and cardiovascular outcomes. Altogether, the psnCVD framework offers a network-based methodology for patient clustering, outperforming that of traditional K-means clustering from raw clinical variables (**S9 Fig**).

## Longitudinal patient–patient network analysis

To further explore network characteristics associated with CTRCD, we performed longitudinal patient–patient network analyses over patient's morbidity and mortality with over 20 years' follow-up data. We tracked the distribution of de novo CTRCD and mortality for all patients across 4 subgroups. Specifically, we inspected 4 consecutive time periods after cancer therapy initiation based on over 20 years' follow-up from our institutional EMRs. From the distribution of de novo CTRCD and mortality, cancer patients with de novo CTRCD were enriched in orange and green subgroups across multiple time points (**Fig 3**). However, patients in subgroups purple and orange show the worse mortality within 10 years of cancer therapy initiation (**Fig 3**), consistent with survival analysis in the combined patient cohort (**Fig 2C**). From the temporal distribution of de novo CTRCD, the patients in orange subgroup had a higher

percentage of de novo CTRCD during the cancer therapy initiation of 0 to 1 year and 2 to 5 years in comparison to long-term exposure (6 to 20 years) (0 to 1 year 10.2% 95% CI 7.8% to 12.6%, 2 to 5 years 11.4% 95% CI 8.8% to 13.8%, 6 to 10 years 4.6% 95% CI 2.9% to 6.2%, and 11 to 20 years 2.2% 95% CI 1.1% to 3.4%, $p < 0.001$; **Fig 3**), suggesting acute cardiotoxicity [44,45]. In addition, we found the worst mortality after the cancer therapy initiation of 2 to 5 years (**Fig 3**), indicating important roles of early cardiac care in improving of cancer patients' survival. For example, 29.2% patients died in purple subgroup during years 2 to 5 in comparison to years 6 to 10 (10.8%) and years 11 to 20 (4.3%) (**Fig 3**).

We further calculated the incidence of 5 types of de novo CTRCD events from chemotherapy initiation date from 1 to 20 years (**S7 Table**). We found that 32% (1-year incidence is 6.11%) of de novo CTRCD events were diagnosed in the first year after chemotherapy, especially for 35% of HF events (1-year incidence is 2.05%) and 36% of AF (incidence is 2.12%) (**S7 Table** and **S10 Fig**). Notably, the 5-year incidence of all 5 de novo cardiovascular events are 13.49% (**S7 Table**), and 71% of cardiovascular events were diagnosed in the first 5 years (**S10 Fig**) during the 20-year follow-up window, further suggesting acute cardio-toxicity and importance of long-term cardiac care for cancer survivors.

## Network-based discovery of clinically actionable variables

We further performed clinical variable–variable network analyses to identify actionable biomarkers for characterization of de novo CTRCD outcomes and mortality rate. Clinical variables were divided into 4 categories: cardiac, echocardiogram, lab testing, and general demographics. A key finding is that cardiac variables (including Troponin T and NT-proB-type Natriuretic Peptide [NT-proBNP]) have a stronger connectivity in the highest de novo CTRCD risk subgroup (orange) compared to the lowest risk subgroup (blue) (**Figs 4** and **5A**). Troponin T [46] and NT-proBNP [47] are 2 well-established cardiac biomarkers for risk assessment of heart disease. Troponin T and NT-proBNP have a stronger betweenness centrality in the highest de novo CTRCD risk subgroup (orange) compared to the other 3 clinical variable networks (**S11A Fig**). Creatinine is another clinical variable with a strong connectivity and centrality in the orange subgroup compared to other 3 subgroups (**Figs 4** and **5A**). Meanwhile, creatinine is highly connected with Troponin T and NT-proBNP in orange, green, and purple subgroups (**Fig 4**). In contrast, creatinine loses connectivity with Troponin T or NT-proBNP in the blue subgroup. These observations suggest a clinical role of creatinine in risk assessment of CTRCD.

Next, we inspected levels of network-predicted biomarkers (**Fig 5A**) in patient data (**S8 Table**). We found that patients had an elevated serum Troponin T (orange mean = 0.15 μg/L 95% CI 0.10 to 0.20 μg/L versus blue mean = 0.03 μg/L 95% CI 0.01 to 0.02 μg/L, $p < 0.001$, KS test; **Fig 5B**) and an elevated serum creatinine (orange mean = 1.08 mg/dL 95% CI 1.01 to 1.34 mg/dL versus blue mean = 0.84 mg/dL 95% CI 0.83 to 0.85 mg/dL, $p < 0.001$, KS test; **Fig 5B**) level in the orange subgroup compared to the lowest CTRCD risk subgroup (blue), consistent with the clinical variable network analysis (**Fig 4**). We found significant changes for several echocardiogram parameters (including LVEF, end-diastolic volume (EDV), and end-systolic volume (ESV)) in orange subgroup compared to blue and purple subgroups ($p < 0.001$, KS test; **Fig 5B** and **S11B Fig, S8** and **S9 Tables**). As expected, several general demographic variables, including BMI and body surface area (BSA), were significantly elevated in orange and green subgroups compared to blue and purple subgroups ($p < 0.001$, KS test; **Fig 5B** and **S11B Fig, S8** and **S9 Tables**). One key finding is that an elevated serum level of NT-proBNP ($p < 0.001$, KS test; **Fig 5B, S8** and **S9 Tables**) and Troponin T ($p < 0.001$, KS test; **Fig 5B, S8** and **S9 Tables**) is observed in both orange (the highest CTRCD risk) and purple subgroups

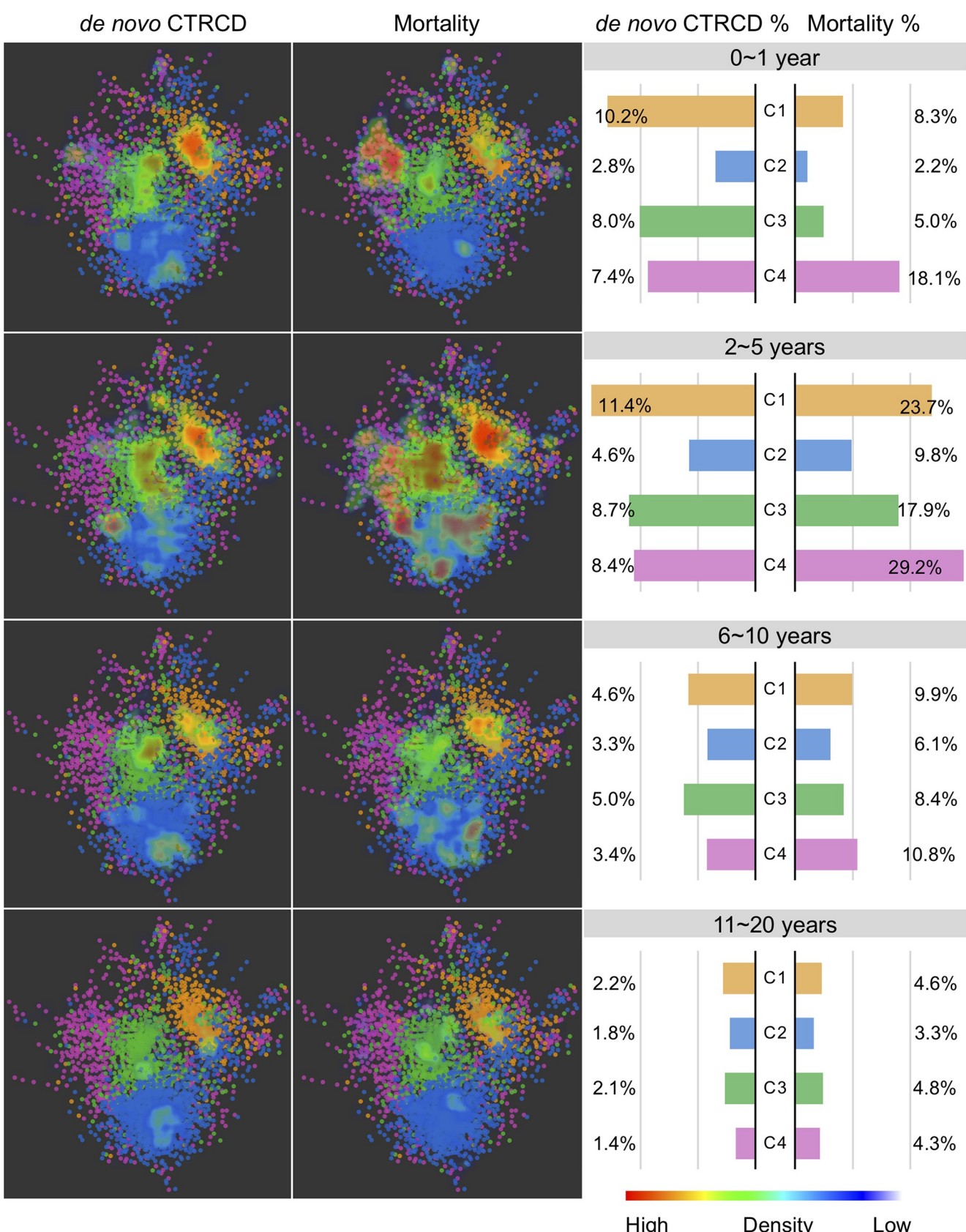

**Fig 3. Longitudinal patient–patient network analysis.** The patient–patient network colorized by cluster numbers with red to blue gradual heat map indicating the de novo CTRCD (left) and mortality (right) distribution in the network. A gradient of red to blue color was used to highlight de novo CTRCD and mortality outcomes among the different patient subgroups (whereby dense red saturation means more enriched outcomes for the patients in that area of the network, and more blue saturation low density of outcomes) across 4 different time points. The right bar plot shows the percentage of de novo CTRCD outcome and mortality across 4 subgroups (Fig 2) during 4 consecutive time periods after cancer therapy initiation. Color key for 4 patient subgroups is consistent with Fig 2. CTRCD, cancer therapy–related cardiac dysfunction.

(the worst mortality) compared to blue subgroup. Importantly, serum levels of NT-proBNP and Troponin T are significantly correlated with patient's mortality ($p < 0.001$, log-rank test; Fig 6). The HR was 2.95 (95% CI 2.28 to 3.82, $p < 0.001$) between NT-proBNP > 900 pg/mL and NT-proBNP = 0 to 125 pg/mL. The HR was 2.08 (95% CI 1.83 to 2.34, $p < 0.001$) between Troponin T > 0.05 μg/L versus Troponin T ≤ 0.01 μg/L. In summary, combining clinical variable network analyses and survival analysis revealed that Troponin T and NT-proBNP offer potential actionable biomarkers for cardiac risk assessment of patients during cancer treatment.

To further confirm the significance of the network-discovered variables, we next turned to perform Cox regression–based HR analyses. Firstly, we computed the PCC among all 112 features to test the collinearity of paired features (S10 Table). As shown in S12 Fig, approximate 95% variable–variable pairs have |PCC| values less than 0.25, suggesting overall low collinearity of the variables. We performed Cox regression model analyses for 22 selected clinical variables having the most connectivity (degree > 10) in the clinical variable network (Fig 4). As shown in Fig 5, the HR analysis is consistent with network-based findings that NT-proBNP and Troponin T are 2 clinically actionable biomarkers for cardiac risk assessment of cancer treatments. To be specific, NT-proBNP and Troponin T are significantly associated with increased risk of de novo CTRCD in orange subgroup (C1; NT-proBNP, HR = 1.36, 95%CI 1.08 to 1.72, $p = 0.010$; Troponin T, HR = 1.139 95%CI 1.00 to 1.30, $p = 0.049$; S11 Table). Meanwhile, the decreased LVEF (parameter from echocardiogram) is significantly associated with increased risk of de novo CTRCD in orange subgroup (HR = 0.96, 95%CI 0.93 to 0.98, $p = 0.003$; S11 Table). Altogether, these HR analyses further confirmed network-based findings.

## Validation of model generalizability

Since our patient clustering method in this study is unsupervised, the common train-test evaluation strategy used in supervised machine learning cannot be applied here directly. We performed random-split and time-split training-test validation strategies to evaluate the generalizability of our psnCVD models. In the time-split, the set with earlier time was regarded as the training set, while in random-split, all patients were randomly split to 2 equally sized training versus test sets by 3 random experiments. We fitted the K-means model on the training set and used the model to predict the clusters for the test set. The detailed diagram of the new experiments is illustrated in S13 Fig. We found that survival and cumulative hazard of de novo CTRCD were significantly distinguishable across 4 subgroups in test sets for both time-split (S14 Fig) and random-split experiments (S15 Fig). We further performed time-dependent area under the receiver operating characteristic curve (AUROC) analysis [48–50]. We found that our psnCVD models can further improve performance of the Cox models (S16 Fig). Altogether, these observations revealed a strong generalizability of psnCVD models, suggesting its potential implications for cardio-oncology patients. Yet, further external validation using independent cohorts from different healthcare systems are highly warranted.

## Discussion

In this study, we proposed a clinically relevant, network-based methodology, psnCVD, for cardiac risk stratification by incorporating large-scale, longitudinal patients' clinical and

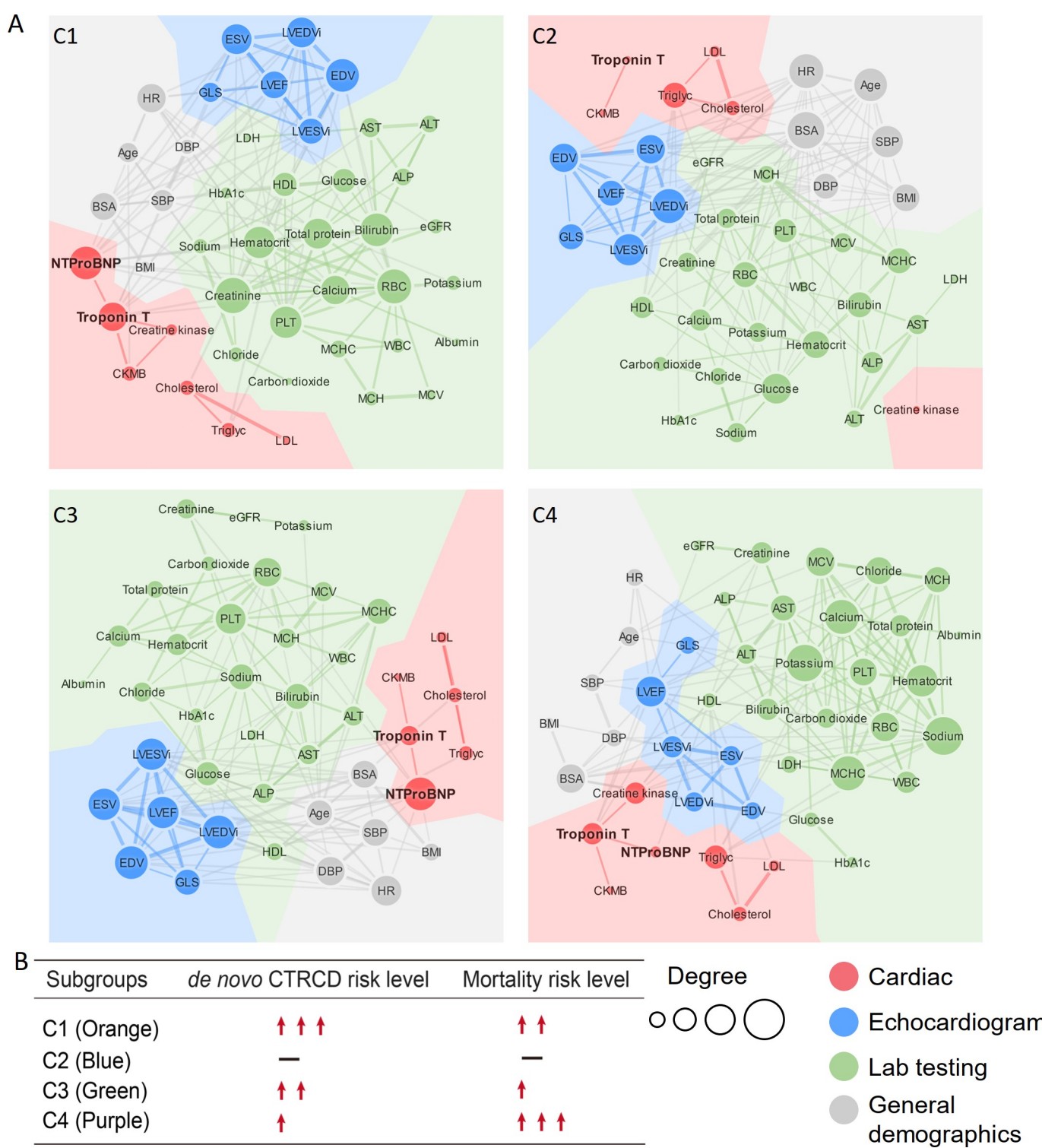

**Fig 4. Clinical variable networks across 4 patient clusters. (A)** Clinical variable–variable networks across 4 patient subgroups: Orange subgroup: C1, intermediate survival and the highest de novo CTRCD risk; blue subgroup: best survival and the lowest de novo CTRCD risk; green subgroup: intermediate survival and intermediate de novo CTRCD risk; purple subgroup: the worst survival and intermediate de novo CTRCD risk. Top 15% of PCC value was used for the final cutoff for each network. At this cutoff, the highest $p$-value among all the correlations in all clusters was $p = 0.008$ (see Methods). Variables were colored by 4 categories of clinical variables: cardiac (red), echocardiogram (blue), lab testing (green), and general demographics (gray). Size of node indicates the degree (connectivity). Size of edges indicates the PCC value in the clinical variable network. **(B)** De novo CTRCD and mortality risk are presented for each subgroup. The abbreviations for all variables are provided in **S1 Table**. CTRCD, cancer therapy–related cardiac dysfunction; NT-proBNP, NT-proB-type Natriuretic Peptide; PCC, Pearson correlation coefficient.

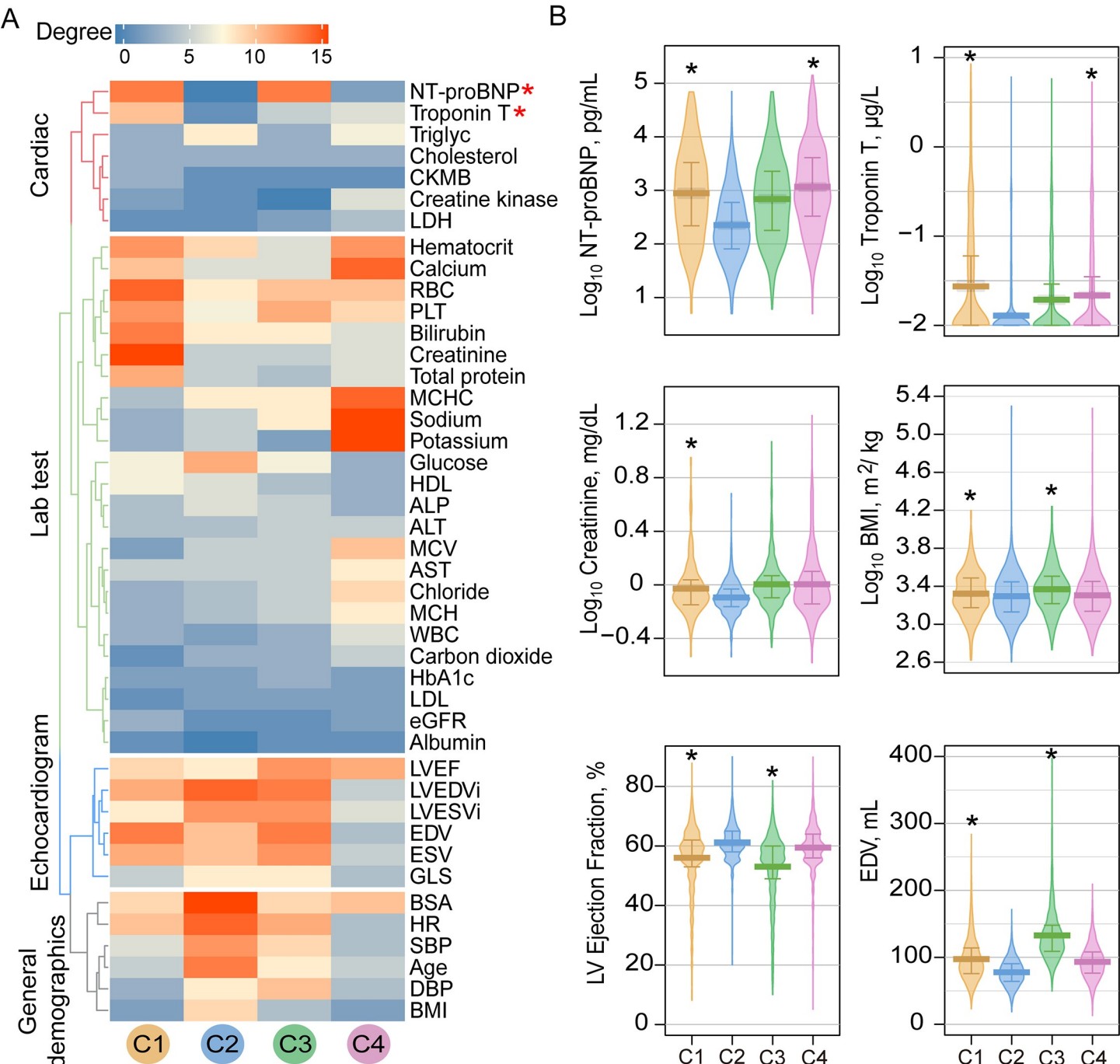

**Fig 5. Network and clinical characteristics of variables across patient subgroups. (A)** Degree distribution of clinical variables across 4 patient subgroup-specific clinical variable networks. The gradient bar shows the degree (connectivity) range. The 4 colored dendrogram indicated 4 types of clinical variables (consistent with **Fig 4A**). The red asterisk highlights the network-identified biomarkers for CTRCD. (**B**) Lab testing values for 6 selected clinical variables across different patient subgroups. The vertical bar denotes the 25% to 75% range, and the thick horizontal lines in each bean plot represent the average value. The black asterisk (*) denotes statistically significantly clinical variables in a specific patient subgroup compared to the C2 subgroup (baseline; **Fig 2**). *p*-value was computed by KS test. All statistical data are provided in **S8 Table**. BMI, body mass index; CTRCD, cancer therapy–related cardiac dysfunction; EDV, end-diastolic volume; KS, Kolmogorov–Smirnov; LVEF, left ventricular ejection fraction; NT-proBNP, NT-proB-type Natriuretic Peptide.

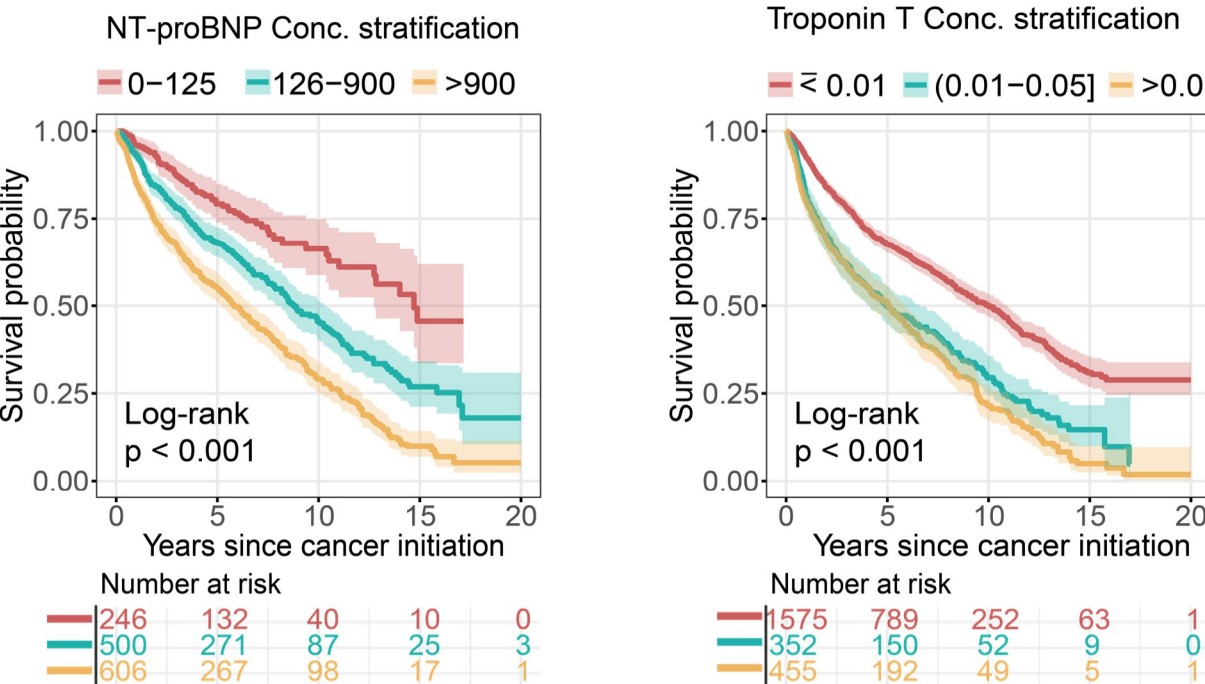

**Fig 6. KM analysis of NT-proBNP and Troponin T in cancer patients.** The threshold of different NT-proBNP (pg/mL) and Troponin T (μg/L) levels were used based on published clinical guidelines. The log-rank test with the BH adjustment [42] was used for survival comparisons among 3 groups. The shadow represents 95% CIs. *p*-value was computed by log-rank test. BH, Benjamini and Hochberg; CI, confidence interval; KM, Kaplan–Meier; NT-proBNP, NT-proB-type Natriuretic Peptide.

echocardiographic data. Using psnCVD, we performed unbiased, network-based analyses of 4,632 cancer patients with 5 diagnosed cardiac outcomes. We identified 4 clinically relevant subgroups of patients using topology-based K-means clustering, including the highest cardiovascular risk group (**Fig 2B**) and the worst mortality group (**Fig 2C**). Importantly, these network-predicted subgroups are significantly correlated with the risk of cardiac dysfunction in cancer survivors during anticancer therapies.

Using longitudinal (up to 20 years' follow-up patient data) patient–patient network analysis, we found that cancer patients have a higher morbidity and mortality within 5 years after the initiation of cancer therapies, indicating acute cardiotoxicity [51]. However, cancer patients have overall low 5-year survival, and the number of patients followed after 10 years is low in our current cohort. Independent cohort validations using EMR-derived time-series patient databases with a longer follow-up time are warranted.

Compared to traditional machine learning–based approaches, network-based approaches are more interpretable, visualizing the decision boundary in the context of topology-based patient–patient networks based on several recent patient network studies [31,52,53]. Previous studies have used unsupervised machine learning method for patient clustering; however, the clinical variables in most published approaches lack clinical interpretation [52]. Using clinical variable network analysis, we found that Troponin-T and NT-proBNP offer potential predictors for assessment of cardiovascular risk in cancer patients (**Figs 4–6**). Our network finding is consistent with a recent meta-analysis that assessment of troponin levels may offer clinical benefits for cancer patients with CTRCD [54]. In addition to cancer-associated cardiotoxicity, CVD is a risk factor for new onset cancer [55]. In clinical variable network-based analysis, we found that cancer patients with the elevated levels of NT-proBNP and Troponin-T had a

worse survival (**Fig 6**), further supporting the potential roles of cardiac biomarkers involved in cancer survival. In addition to NT-proBNP and Troponin-T, several lab testing variables, such as sodium and potassium, have high connectivity in patients within the worst mortality subgroup (purple), revealing potential prognostic markers in cancer patients with cardiovascular events (**Fig 5**, **S11B Fig**). Altogether, assessment of troponin levels or other serum markers may qualify as a screening test to identify patients who require referral to cardio-oncology units and benefit from preventive strategies of cardiovascular risk. Further independent validation and clinical trials are warranted before used as biomarkers in clinics.

We acknowledge several limitations. We used the ICD 9/10 codes to define 5 types of cardiovascular events before and after cancer treatments. The accuracy of ICD 9/10 codes may influence possible false positive findings during cardiovascular outcome validations. Risk estimates may have been subject to bias within individuals because of variability in patient referral pattern, clinical volume, and threshold for hospitalization in the EMR database. Patient–patient similarity may be nonlinear, which cannot be measured by a linear measure. In this study, we adjusted various confounding factors, including age, cardiac risk factors (diabetes and hypertension), family history, and others based on our sizeable efforts. Yet, other possible confounding factors, including disease comorbidities, multiple medication usages (such as combination regimens among radiotherapy, chemotherapy, and targeted therapy), and others, may influence our findings. Although we found that other confounding factors, such as tumor stages, tumor type, and anticancer medications, have minor impacts on patient network-based findings (**Fig 2, S8 Fig**), further confounding factor adjustment tested in other independent cohorts are needed in the future. To inspect influence of heterogeneities of anticancer medications, we rebuilt a patient–patient network using a subpopulation of patients ($n = 1,252$) who received chemotherapy only (**S17A Fig**). Utilizing psnCVD framework, we identified 3 clinically relevant subgroups in this small, homogeneous population: Cardiovascular outcomes ($p < 0.001$, log-rank test; **S17B Fig**) and patient survival rate ($p < 0.001$, log-rank test; **S17C Fig**) are highly correlated with patient subgroups as well. In this study, we used a K-means clustering approach that may overfit for network-based patient clustering [56]. We observed high overall performance (**S13**–**S15 Figs**) and a strong generalizability (**S16 Fig**) of psnCVD models using random-split and time-split training-test validation strategies. In addition, psnCVD models improve the performance of the Cox proportional hazard models during time-dependent AUROC analysis [48–50]. These observations indicate a strong generalizability of our psnCVD methodology. However, additional prospective studies in different healthcare systems and EMR databases are highly warranted to validate the generalizability of psnCVD models before clinical use. Finally, the development of an online risk calculator by integrating all patient–patient network models would provide useful tools for cardiac risk assessment during cardio-oncology clinical practices. For example, to permit an unbiased risk stratification for new individuals, the clinical variables of individuals can be collected by research electronic data capture (REDCap) tools during the cardio-oncology practices. The cluster of a new patient will be predicted based on the collected clinical variables using our psnCVD models [26,29].

In summary, this study implies that an unbiased, systems-based network analysis of large-scale, longitudinal patient data is more interpretable, visualizing the decision boundary to cardiac risk stratification for patients before, during, and after cancer treatment. Importantly, the network methodologies will excel at integrating heterogeneous patient data and generating interpretable, clinical insights of models. From a translational perspective, if broadly applied, the network tools developed here hold great promise for identifying novel cardiac risk subgroups and clinically actionable biomarkers for rapid development of precision cardio-oncology.

## Supporting information

**S1 Checklist. STARD Checklist.**
(PDF)

**S1 Fig. KM curves to estimate the survival and cardiovascular outcome for different number of clusters.** The number of clusters represents different K values ranging from 3 to 10 in K-means clustering. The log-rank test was used to evaluate the statistical significance. All pairwise *p*-values between the subgroups for each K value were summarized in S2 Table. CVD, cardiovascular disease; KM, Kaplan–Meier.
(PDF)

**S2 Fig. Clustering stability test.** (**A**) The workflow of K-means clustering stability test. (**B**) The ARI and AMI among the 100 repeats showed high stability of the clustering results. The averages and standard deviations are shown in the bar plot. AMI, adjusted mutual information; ARI, adjusted rand index.
(PDF)

**S3 Fig. Network density–based cosine cutoff selection.** The network density at different cutoff values and selected the cutoff that resulted in the lowest network density. Network density is defined as the ratio of the number of actual links and the number of all possible links from all the patients.
(PDF)

**S4 Fig. PCC as patient similarity metric.** (**A**) Patient–patient network colorized by 4 cluster numbers. All edges have PCC < 0.65 for the patient pairs. All data preprocessing and PCC cutoff selection were same with the method cosine similarity calculation. The network was visualized using Cytoscape v 3.7.1. (**B**) KM curves to estimate the all-cause survival probability in the 4 subgroups. The log-rank test was used to evaluate the statistical significance. KM, Kaplan–Meier; PCC, Pearson correlation coefficient.
(PDF)

**S5 Fig. Variable network PCC cutoff selection.** 5%, 10%, 15%, and 20% were used to test the K% connections with the highest PCC for the construction of the network. PCC, Pearson correlation coefficient.
(PDF)

**S6 Fig. Efficacy of the risk stratification on each CVD outcomes.** (**A**) Cumulative hazard of 5 de novo CVD events across 4 subgroups are shown. The log-rank test with the BH adjustment was used for comparing the cumulative hazard among 4 subgroups. The shadow represents 95% CI. (**B**) The percentage of 5 CVD events across 4 subgroups. (**C**) The percentage of 5 de novo CVD events (the patient has at least one type of cardiac event diagnosed after cancer therapy) across 4 subgroups. AF, atrial fibrillation; BH, Benjamini and Hochberg; CAD, coronary artery disease; CI, confidence interval; CVD, cardiovascular disease; HF, heart failure; MI, myocardial infarction.
(PDF)

**S7 Fig. HR of mortality across 4 subgroups.** HRs (and 95% CI) of CTRCD, cancer type, and cancer stage aim to mortality outcome. The Wald $\chi^2$ test was used to evaluate the variables with statistically significant coefficients. CI, confidence interval; CTRCD, cancer therapy–related cardiac dysfunction; CVD, cardiovascular disease; HR, hazard ratio.
(PDF)

**S8 Fig. Outcome validation for risk stratification model on the clinically derived variables plus cancer type, cancer stage, and treatment type.** (**A**) KM curves to estimate all survival probability and (**B**) cumulative hazard of de novo CTRCD (the patient has at least one type of cardiac event diagnosed after cancer therapy) across 4 subgroups are shown. The log-rank test with the BH adjustment was used for comparing the cumulative hazard among 4 subgroups. The shadow represents 95% CI. BH, Benjamini and Hochberg; CI, confidence interval; CTRCD, cancer therapy–related cardiac dysfunction; KM, Kaplan–Meier.
(PDF)

**S9 Fig. Outcome validation for K-means clustering directly on the clinically derived variables for 4,632 patients.** (**A**) KM curves to estimate all survival probability across 4 subgroups are shown and (**B**) cumulative hazard of de novo CTRCD (the patient has at least one type of cardiac event diagnosed after cancer therapy). The log-rank test with the BH adjustment was used for comparing the cumulative hazard among 4 subgroups. The shadow represents 95% CI. BH, Benjamini and Hochberg; CI, confidence interval; CTRCD, cancer therapy–related cardiac dysfunction; KM, Kaplan–Meier.
(PDF)

**S10 Fig. Cumulative percentage of 5 de novo CTRCD events from chemotherapy initiation 1 year, 5 years, 10 years, and 20 years.** AF, atrial fibrillation; CAD, coronary artery disease; CTRCD, cancer therapy–related cardiac dysfunction; HF, heart failure; MI, myocardial infarction.
(PDF)

**S11 Fig. Betweenness centrality of the variables.** (**A**) Betweenness centrality of clinical variables across 4 patient subgroup-specific clinical variable network. The gradient bar shows the centrality range. (**B**) Lab testing values for 4 selected clinical variables across different patient subgroups. The vertical bar denotes the 25% to 75% range, and the thick horizontal lines in each bean plot represent the average value. The black asterisk (*) denotes statistically significantly clinical variables in specific patient subgroup compared to the C2 subgroup. *p*-value was computed by KS test. All statistical data are provided in **S8 Table**. BSA, body surface area; ESV, end-systolic volume; KS, Kolmogorov–Smirnov.
(PDF)

**S12 Fig. Pairwise Pearson correlations among the used 112 clinical variables.** The gradient red color denotes positive correlation, and gradient blue color denotes negative correlation. The order of labels in heatmap were followed by 4 variable categories. Due to the space limitation, the labels in heatmap show one name in every 3 names. The full correlation matrix of 112 variables were showed in **S10 Table**, and the order of variable labels were the same with the label ranked in the heatmap.
(PDF)

**S13 Fig. The workflow of the train-test validation strategy to evaluate the generalizability of psnCVD models.** All patients were split randomly or by time to training and test sets. We computed the cosine similarity matrix for patients in the training set (blue matrix) and for patients in the test set (green matrix) against the training set. Next, the K-means clustering was performed on the training set and was used to predict both the training and test sets. The predicted clusters were evaluated for the survival and de novo CTRCD risk for both the training and test sets. CTRCD, cancer therapy–related cardiac dysfunction; psnCVD, patient–patient similarity network-based risk assessment of CVD.
(PDF)

**S14 Fig. Evaluation of the generalizability of psnCVD models using time-split cohorts.**
Patients were split by their cancer diagnosis time to 3 training set/test set pairs: 50% versus
50%, 60% versus 40%, and 80% versus 20%, respectively. The survival probability and cumulative hazard of de novo CTRCD of the training sets and test sets were evaluated. Log-rank tests
show statistically significant difference in survival probability and cumulative hazard of de
novo CTRCD for the patient groups in the test sets. CTRCD, cancer therapy–related cardiac
dysfunction; psnCVD, patient–patient similarity network-based risk assessment of CVD.
(PDF)

**S15 Fig. Evaluation of the generalizability of psnCVD models using randomly split
cohorts.** The survival probability and cumulative hazard of de novo CTRCD of the training set
(50%) and test set (50%) were evaluated in 3 independent random experiments. Log-rank tests
show statistically significant difference in survival probability and cumulative hazard of de
novo CTRCD for the patient groups in the test sets. CTRCD, cancer therapy–related cardiac
dysfunction; psnCVD, patient–patient similarity network-based risk assessment of CVD.
(PDF)

**S16 Fig. Time-dependent AUROC analysis of Cox proportional hazard models using the
entire cohort (A) and individual patient subgroups identified by psnCVD models (B–E).**
The overall performance of Cox proportional hazards model using the entire cohort (**A**) and
individual patient subgroups (**B–E**). For each subplot, all patients (A) or patients in individual
subgroups (**B–E**) were randomly split to training (50%) and test (50%) set. The clusters for the
patients in the test set were predicted based on the model fitted on the training set. Time-
dependent AUROC was used to evaluate the model performance of the test sets. AUROC, area
under the receiver operating characteristic curve; psnCVD, patient–patient similarity net-
work-based risk assessment of CVD.
(PDF)

**S17 Fig. Methodology application in chemotherapy population.** (**A**) Patient–patient net-
work colorized by 3 cluster numbers. Patient–patient network using a subpopulation of
patients ($n = 1,252$) who received chemotherapy only. Using cosine $< 0.55$ as a cutoff, 3 clus-
ters were identified: cluster 1a ($n = 502$), cluster 2a ($n = 474$), and cluster 3a ($n = 275$). The net-
work was visualized using Cytoscape v3.7.1. (**B**) Cumulative hazard of de novo CTRCD in the
3 subgroups. The log-rank test was used to evaluate the statistical significance. (C) KM curves
to estimate the all-cause survival probability in the 3 subgroups. CTRCD, cancer therapy–
related cardiac dysfunction; KM, Kaplan–Meier.
(PDF)

**S1 Table. The full information of 112 clinical variables used in this study.**
(XLSX)

**S2 Table. Summary of survival and cardiovascular outcome validations across different
number of clusters.**
(XLSX)

**S3 Table. Baseline characters and clinical outcomes of orange (C1) subgroup.**
(XLSX)

**S4 Table. Baseline characters and clinical outcomes of green (C3) subgroup.**
(XLSX)

**S5 Table. Baseline characters and clinical outcomes of blue (C2) subgroup.**
(XLSX)

**S6 Table. Baseline characters and clinical outcomes of purple (C4) subgroup.**
(XLSX)

**S7 Table. The incidence of 5 de novo cardiovascular outcomes from cancer therapy initiation across 20 years.**
(XLSX)

**S8 Table. Statistics analysis of clinical variable across 4 subgroups.**
(XLSX)

**S9 Table. Summary of clinically actionable variables.**
(XLSX)

**S10 Table. The correlation matrix for 112 clinical variables.**
(XLSX)

**S11 Table. The hazard ratio analysis for the selected clinical variables.**
(XLSX)

## Author Contributions

**Conceptualization:** Patrick Collier, Feixiong Cheng.

**Data curation:** Yuan Hou, Yadi Zhou, Muzna Hussain, G. Thomas Budd, Wai Hong Wilson Tang, James Abraham, Bo Xu, Chirag Shah, Rohit Moudgil, Zoran Popovic, Chris Watson, Leslie Cho, Mina Chung, Mohamed Kanj, Samir Kapadia, Brian Griffin, Lars Svensson, Patrick Collier, Feixiong Cheng.

**Formal analysis:** Yuan Hou, Yadi Zhou, Muzna Hussain, Feixiong Cheng.

**Funding acquisition:** Feixiong Cheng.

**Investigation:** Yuan Hou, Feixiong Cheng.

**Methodology:** Yuan Hou, Feixiong Cheng.

**Project administration:** Feixiong Cheng.

**Resources:** G. Thomas Budd, James Abraham, Zoran Popovic, Leslie Cho, Samir Kapadia, Brian Griffin, Lars Svensson, Patrick Collier, Feixiong Cheng.

**Software:** Yuan Hou, Feixiong Cheng.

**Supervision:** Patrick Collier, Feixiong Cheng.

**Validation:** Yuan Hou, Yadi Zhou, Muzna Hussain, Patrick Collier, Feixiong Cheng.

**Visualization:** Yuan Hou, Feixiong Cheng.

**Writing – original draft:** Yuan Hou, Yadi Zhou, Muzna Hussain, Feixiong Cheng.

**Writing – review & editing:** Patrick Collier, Feixiong Cheng.

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
