## [Editor Report · Decision Letter 0]

17 Feb 2020

Dear Dr Cheng, 

Thank you for submitting your manuscript entitled "Longitudinal Population-based Cardiac Risk Stratification in 4,600 Cancer Patients from 1997 to 2019" for consideration by PLOS Medicine.

Your manuscript has now been evaluated by the PLOS Medicine editorial staff and I am writing to let you know that we would like to send your submission out for external peer review.

Kind regards,

Helen Howard, for Clare Stone PhD 

Acting Editor-in-Chief

PLOS Medicine 

plosmedicine.org

---

## [Decision Letter · Decision Letter 1]

4 Aug 2020

Dear Dr. Cheng,

Thank you very much for submitting your manuscript "Longitudinal Population-based Cardiac Risk Stratification in 4,600 Cancer Patients from 1997 to 2019" (PMEDICINE-D-20-00473R1) for consideration at PLOS Medicine. 

Your paper was evaluated by a senior editor and discussed among all the editors here. It was also evaluated by three independent reviewers, including a statistical reviewer. The reviews are appended at the bottom of this email and any accompanying reviewer attachments can be seen via the link below:

[LINK]

In light of these reviews, I am afraid that we will not be able to accept the manuscript for publication in the journal in its current form, but we would like to consider a revised version that addresses the reviewers' and editors' comments. Obviously we cannot make any decision about publication until we have seen the revised manuscript and your response, and we plan to seek re-review by one or more of the reviewers. 

We expect to receive your revised manuscript by Aug 25 2020 11:59PM. Please email us (plosmedicine@plos.org) if you have any questions or concerns.

We look forward to receiving your revised manuscript. 

Sincerely,

Emma Veitch, PhD

PLOS Medicine

On behalf of Clare Stone, PhD, Acting Chief Editor,

PLOS Medicine

plosmedicine.org

*We'd suggest revising the title according to PLOS Medicine's style, ideally this should include an indication of the study design (eg, "A randomized controlled trial," "A retrospective study," "A modelling study," etc.) in the subtitle (ie, after a colon).

*In the last sentence of the Abstract Methods and Findings section, please include a brief note about any key limitation(s) of the study's methodology.

*We would ask that the authors clarify in the paper whether the analytical approach reported here corresponds to one laid out in a prospective protocol or analysis plan? Please state this (either way) early in the Methods section.

*If appropriate, the authors could consider using the TRIPOD guideline (https://www.equator-network.org/reporting-guidelines/tripod-statement/) to support reporting of their study. 

Comments from the reviewers:

Reviewer #1: "Longitudinal Population-based Cardiac Risk Stratification in 4,600 Cancer Patients from 1997 to 2019" introduces a network clustering approach on longitudinal clinically-derived variables, to identify four clinically-relevant subgroups with correlations to cancer therapy-related cardiac dysfunction (CTRCD) outcomes. Analysis of the network further identified particular cardiac variables that may be actionable biomarkers associated with CTRCD.

The use of clustering methods in grouping patients has been popular in medicine, likely due to the natural intuition that if two patients have similar profiles as far as is known, their outcomes should also be similar, or at least more than between patients with dissimilar profiles. The availability of the relevant code (with accompanying commentary) on GitHub was appreciated for aiding understanding.

However, there remain two relatively major concerns that might be addressed:

Firstly, for this manuscript, a patient-patient similarity network was constructed before (K-means) clustering was employed. The cosine similarity metric (Equation 1; also commonly seen in evaluating embedding distances) was employed to determine whether two patients are sufficiently-similar (i.e. have cosine similarity above some cutoff), and thus connected in the similarity network. While not explicitly stated, it is implied from S1 Fig that if a particular patient has no similar-enough fellows, that patient is not included in the similarity network (because as the cutoff threshold increases, the number of patients/nodes in the network decreases with increasing number of disconnected nodes)

1a. It is not explicitly explained why a similarity network with minimized network density (i.e. as few links as possible, for a given maximum possible number of links as determined by the number of nodes in the network) is desirable. Intuitively, links between nodes in a network with low density might be considered more "meaningful", since they are relatively rare. However, the tradeoff (as seen in S1 Fig) is that a proportion of patients would have no links due to the cutoff threshold, and will thus be excluded from all subgroups (from Page 10, the final network contains only 3,131 of the original 4,632 patients). The authors might consider more formally explaining the choice of minimized network density, with references if appropriate.

Further on minimizing network density, there appears to be no guarantee that a cutoff that yields minimum density would also retain a meaningful number of patients/nodes (e.g. from S1 Fig, fewer than 1000 patients/nodes remain at a cutoff of 0.75). Does this possibility factor into the choice of the network density metric?

1b. It is then stated that patients were clustered "using their network profiles". "Network profile" does not seem to be defined in the text, but from the GitHub, it appears that a patient's profile consists of all other similar patients (by cosine similarity), but without the similarity value (i.e. once above the cutoff, it is not considered whether the patients are barely sufficiently similar, or essentially identical). The definition of network profile might be provided in the text.

1c. Following from 1a & 1b, it is unclear why the similarity network is necessary. The K-means clustering could conceivably have been directly performed on the clinically-derived variables for all patients, as would seem the usual practice. This would moreover not detract from interpretability, since all patients would remain as nodes in a topological space. The authors might cite/describe theoretical support for psnCVD, and/or empirical evidence that it is superior to direct K-means clustering as a baseline.

1d. The x-axes for S1 Fig read "PCC cutoff", but Pearson correlation coefficient (PCC) appears to be an alternative metric to cosine similarity, which is supposed to be used in S1 Fig. Is this a mislabeling?

1e. On Page 10, it is stated that "we tested the cutoffs in an increment of 0.5". Might this be "...in increments of 0.05" instead?

1f. Given that a cutoff of 0.65 was identified as resulting in minimum network density, why was the connection cutoff between patients then states ad (> 0.62) in the next sentence, rather than 0.65?

Secondly, there may be concerns with the replicability of the findings:

2a. K-means clustering remains a machine learning technique that possibly overfits given data. As such, to ensure that clinical subgroups discovered via clustering are reliable, common practice is to reproduce the clustering on independent cohorts (e.g. in "Novel subgroups of adult-onset diabetes and their association with outcomes: a data-driven cluster analysis of six variables", Ahlqvist et al., The Lancet Diabetes & Endocrinology, 2018), or on held-out validation data.

2b. K-means clustering may moreover have a stochastic component, resulting in somewhat different clusters being produced from the same input data. If so, the clusterwise stability might be assessed (also as in Ahlqvist et al.)

Other comments follow:

3. The nature of the clinical variables used is not entirely clear. In particular, it is stated on Page 9 that "we obtained 112 variables (including the derived ones). A detailed description for all the variables can be found in the supplemental methods section"; however, the exact figure/table listing these 112 variables does not appear to be present in the supplementary material. The closest appears to be S1 Table, which lists the abbreviation of some of the variables. A complete list and description of these variables might be added.

4. While the greater interpretability of a network-based approach is cited as a motivation for using a network rather than other machine learning techniques, relatively-interpretable statistical methods such as logistic regression would appear appropriate for identifying potential biomarkers/clinically actionable variables. The authors might consider confirming the significance of the network-discovered biomarkers/variables independent of subgroup analysis (e.g. with hazard ratios).

5. The CTRCD outcome currently consists of five cardiac conditions/events lumped together. It could be interesting to comment on the efficacy of the risk stratification on each of these conditions.

6. There remain some minor grammatical issues [e.g. "...each patient totally had 56 clinical variables" (Page 8), "...from all cause" (Page 12) "...shows the worse mortality" (end of Page 15), "...have the limited clinical variables" (Page 19), "Total cohort were 4.532 patients" (Page 31)] and possible spelling issues [e.g. "K-mean" instead of "K-means" (Page 3/18)]

Reviewer #2: Reviewer's Comments for PMEDICINE-D-20-00473

Authors aimed to perform unbiased cardiac risk stratification for cancer patients using a single institutional electronic medical record (EMR). The study covers an interesting issue, however there are several important concerns. 

Comments

1. In this study, the cardiovascular (CV) event and mortality were confirmed using their own EMR system. How did authors analyze the outcomes for patients who were initially treated in Cleveland Clinic but moved to another institution? What is the percentage of patients whose CV events or deaths are unknown for over six months? Validity of the current study significantly relies on the quality of CV event verification, specifically the accuracy of their ICD codes. Is there any external validation study for their ICD code based diagnosis for atrial fibrillation, coronary artery disease, heart failure, myocardial infarction and stroke? 

2. Lack of external validation for the current classification is an important limitation. If external validation is not possible, the whole study population could be randomly divided into derivation cohort and validation cohort for internal validation. Phenotype based classification without independent validation poses a significant potential for bias.

3. Not only the types and stages of specific cancer, but treatment information such as cumulative doses of anthracycline, radiation therapy or specific targeted agent confer a higher cardiotoxic effect. However, current risk stratification model does not include any information regarding types, stages or treatment of specific cancer. 

4. Authors should mention about concrete application methods of their findings. If a new cancer patient is referred to cardio-oncology clinic, how can this risk stratification method can be applied? 

Reviewer #3: Hou et al. have performed cardiac risk stratification for cancer patients using patient-patient network analysis. The filed of cardio-oncology is emerging rapidly and creating a risk stratification of the patients in risk for developing cardiotoxicity is extremly important as it may allow the begining of cardioprotective therpay and prevent the interuption of cancer therpay. 

The use of netwrok based methodology is intresting and innovative. I have some minor comments:

1. The term CTRCD is usually accepted for LVEF reduction >10% according to the ESC / ASE/ EACVI..... For my understaing in this paper CTRCD was considered as AF / MI/ STROKE/ HF/ CAD, which may be confusing. Therfore I would condiser to change the term CTRCD to cardiovascular events. 

2. Do you have any information reagrding the incidence of each event for the cardiocvascular events? AF ? MI ?....

3. Can you explain why you choose to include AF with MI/STROKE/HF/CAD? The incidence and the severity of AF is not equivalent to the other events and I'm not sure that should be included in teh same category. Most probably cancer therapy will not be interapted due to AF as it will be due to MI or HF. 

4. The outcome of the paper included cardiovascular events and all-cause mortality. Do you have any information regarding CV mortality which is more relevent to cardiotoxicity since probably the majority of the death were due to cancer reasons and not relates to cardiotoxicity. 

5. Regarfing the observation according to years - I'm not sure that we can conclude that the high mortality in 5 years imply the dose time dependent since cancer patients has low 5 years survival, and furthermore the number of patients followed after 10 years is vey low.

[LINK]

---

## [Decision Letter · Decision Letter 2]

23 Oct 2020

Dear Dr. Cheng,

Thank you very much for submitting your revised manuscript "Longitudinal Population-based Cardiac Risk Stratification in 4,600 Cancer Patients from 1997 to 2019" (PMEDICINE-D-20-00473R2) for consideration at PLOS Medicine. 

Your revision was evaluated by a senior editor and discussed among all the editors here. It was also discussed with an academic editor with relevant expertise, and sent to the statistical reviewer. The reviews are appended at the bottom of this email and any accompanying reviewer attachments can be seen via the link below:

[LINK]

As you will note below, the statistical reviewer still has strong concerns. In addition to the remaining points from the statistical reviewer, the Academic Editor notes that it is not clear how the predictive performance of this model compares with those of a standard regression model. In light of these comments, I am afraid that we still will not be able to accept the manuscript for publication in the journal in its current form, but we would like to consider a revised version that addresses the reviewers' and editors' comments. Obviously we cannot make any decision about publication until we have seen the revised manuscript and your response, and we plan to seek re-review. 

We expect to receive your revised manuscript by Nov 13 2020 11:59PM. Please email us (plosmedicine@plos.org) if you have any questions or concerns.

We look forward to receiving your revised manuscript. 

Sincerely,

Thomas McBride, PhD

Senior Editor 

PLOS Medicine

plosmedicine.org

Comments from the Academic Editor:

One issue that I didn't see raised by any of the other reviewers is the overall model performance. If I understand correctly the authors divide their work into three main components: clustering, risk prediction, and a sort of causal inference to discover modifiable risk factors. The first bit is strong. The second component (risk prediction) is done in each of the main clusters separately and HRs compared. Normally, what we would like to also see is the overall model performance (e.g., AUROC/AUPRC) in particular in comparison with conventional models. So, does the modelling add much to a simple Cox model that is applied to all predictors (paying attention to collinearity etc as we would usually do)? The comparison of HRs does not suggest that the clusters are hugely different in risk patterns and the 'unbiased' discovery of modifiable risk factors does not lead to any surprises. Please compare the performance of the predictive model with a standard regression/hazard model.

1- Did your study have a prospective protocol or analysis plan? Please state this (either way) early in the Methods section.

2- Please ensure that the study is reported according to the STARD 2015 reporting guideline for diagnostic accuracy studies, and include the completed STARD checklist as Supporting Information. Please add the following statement, or similar, to the Methods: "This study is reported as per the STARD 2015 reporting guideline for diagnostic accuracy studies (S1 Checklist)."

The STARD guideline can be found here: http://www.equator-network.org/reporting-guidelines/stard/

If you feel a different checklist is more appropriate, please include that instead.

3- Please revise your title according to PLOS Medicine's style. Your title must be nondeclarative and not a question. It should begin with main concept if possible. Please place the study design ("A randomized controlled trial," "A retrospective study," "A modelling study," etc.) in the subtitle (ie, after a colon).

4- In the Abstract Methods and Findings, please include the population and setting, and years during which the study took place.

5- In the Abstract and throughout, please include p-values alongside 95% CIs for all comparisons.

6- In the last sentence of the Abstract Methods and Findings section, please describe the main limitation(s) of the study's methodology.

7- Please make sure all results are first reported in the Results section, rather than the Discussion, which should be focused on interpretation.

Comments from the reviewers:

Reviewer #1: We thank the authors for addressing most of the points raised in the previous review round.

1. On the psnCVD clustering, there might have been slight confusion on the "actual" network after clustering that is used for assigning patients (supposedly using all 4632 patients, described in the Network clustering section), and the "visualization" network that has some less-similar patients removed (and which has minimized network density).

To the best of our updated understanding, the "visualization" network does not actually have any impact on the main results as reported in the Network-based discovery of novel cardiac risk subgroups section; these subgroup results are obtained from the "actual" psnCVD, which moreover can assign a subgroup to any new patient (after that patient's network profile is computed), even if the new patient is actually dissimilar to almost all other patients. As such, the psnCVD can always assign a new patient (possibly from another source/hospital) to a subgroup, given that patient's data.

If this is correct, the authors might consider emphasizing this interpretation, since it was not entirely clear that the visualization density minimization procedure does not actually have an impact on the results.

2. Under the Network clustering section, it is stated that "we instead examining the survival analysis and cardiovascular outcome analyses at different number of clusters. The highest number that produced clusters with distinguishable survival and cardiovascular outcome was 4". How was "distinguishable" determined in this case, i.e. was there some quantifiable metric, or was it by observation?

3. It would be helpful if the main characteristics of the four discovered subgroups (as described in detail in the Network-based discovery of clinically actionable variables section) be summarized within a table.

4. The lack of an independent validation cohort remains a major concern, though this has been mitigated to an extent by two internal validation approaches. More details might be provided on what was done for these internal validation approaches, though:

For 1), the subgroups appear to be split by follow-up time windows (1997-2012), (1997-2015), (1997-2017); was the approach used some form of cross-validation (e.g. to cluster patients from subgroup (1997-2017), the psnCVD was constructed based on the other two subgroups)?

For 2), similarly, given a sampling ratio of 50%, was the psnCVD constructed based on 50% of the randomly-split data, and then evaluated on the remaining 50% to produce the charts?

5. There remain some minor grammatical issues, e.g.

Page 10: "we instead examining the survival analysis..."

Page 11: "we repeated 100 times of the K-means clustering analyses..."

Page 14: "Totally, 1,670 (36%) of patients have at least one type of diagnosed cardiac events..."

Page 17: "long-time exposure..." (long-term?)

[LINK]

---

## [Decision Letter · Decision Letter 3]

7 Jun 2021

Dear Dr. Cheng,

Thank you very much for re-submitting your manuscript "A Longitudinal Patient-Patient Network Analysis of Cardiac Risk in 4,600 Cancer Patients from 1997 to 2019" (PMEDICINE-D-20-00473R3) for consideration at PLOS Medicine. We do apologize for the long delay in sending you a response. 

I have discussed the paper with our academic editor and it was also seen again by one reviewer. I am pleased to tell you that, provided the remaining editorial and production issues are fully dealt with, we expect to be abe to accept the paper for publication in the journal.

[LINK]

Please let me know if you have any questions in the meantime, and we look forward to receiving the revised manuscript shortly.   

Sincerely,

Richard Turner PhD

rturner@plos.org

Requests from Editors:

Please adapt the title to better match journal style. We suggest: "Cardiac risk stratification in cancer patients: A longitudinal patient-patient network analysis".

We suggest substituting "echocardiogram" for "Echo" throughout. 

Please trim the "Background" subsection of your abstract, aiming to remove 1-2 sentences. 

Please quote summary demographic characteristics for study participants in your abstract. 

We note that you quote some very small p values in the abstract. We generally ask for p values to be quoted exactly or as p<0.001, unless there is a specific statistical reason for very small values to be stated exactly. We are not aware of such reasons in this case, and ask you to observe this convention throughout the paper.

Please remove the sentence beginning "However, further prospective validation studies ..." from your abstract. You may wish to make this point as part of a new sentence addressing study limitations (see immediately below) or move it to the Discussion section. 

Please add a new final sentence to the "Methods and findings" subsection of your abstract. This should begin "Study limitations include ..." or similar and should list 2-3 of the study's main limitations. 

Please adapt the "Conclusions" subsection of your abstract to begin "In this study, we found that ..." or similar and adapt the tense(s) used as needed. 

In the "Author summary", please add a few words to the first point of the "What did the researchers do ..." subsection to briefly describe your methodology.

Please restructure the end of the Introduction section of your main text. The final paragraph should briefly state the aim of the study, but not summarize the conclusions. 

Noting "... our [model] excels ..." we ask you to adapt the language used in places to avoid an impression of exaggeration. 

Early in the Methods section (main text), you mention a "retrospective plan". Please adapt the wording here to state that there was no prespecified analysis plan (assuming this is the case - if not, please attach the plan or protocol as a supplementary document, referred to in the text). You may wish to note that the analyses were prespecified and that no data-driven changes were made.

Early in the Discussion section, we suggest rewording "patients' mortality and subsequent cardiac outcomes.".

Throughout the text, please adapt reference call-outs to remove spaces from within the square brackets (e.g., "... ventricular dysfunction [15,16].").

Please use the general style "...4 categories ..." consistently throughout the paper, although numbers should be spelt out at the start of sentences. 

Please remove the information on data availability, funding and competing interests from the end of the main text. In the event of publication, this information will appear in the article metadata via entries in the submission form. 

In table 1 and any other instances in the ms, please substitute "sex" for "gender" where appropriate. 

Please rename the attached checklist "S1_STARD_Checklist" and refer to it by this label in the Methods section. 

Please adapt the checklist so that individual items are referred to by section (e.g., "Methods") and paragraph number, not by line or page numbers (which generally change upon publication). 

"Label" is misspelt in S1 table.

Comments from Reviewers:

*** Reviewer #1: 

We thank the authors for responding to our previous comments, and in particular the additional experiments presented towards validating robustness.

However, while the new S12 Fig is captioned as "Pairwise Pearson Correlations among the used 112 clinical variables", there appear only 38 variables shown in the figure. It is recognized that it may be impractical to display a full matrix of 112 labels, but in this case the caption of the figure might be updated, and commentary made about whether the displayed variables are representative.

***

[LINK]

---

## [Editor Report · Decision Letter 4]

15 Jul 2021

Dear Dr Cheng, 

On behalf of my colleagues and the Academic Editor, Dr Rahimi, I am pleased to inform you that we have agreed to publish your manuscript "Cardiac risk stratification in cancer patients: A longitudinal patient-patient network analysis" (PMEDICINE-D-20-00473R4) in PLOS Medicine.

Prior to final acceptance, please ensure that tenses are used consistently in the abstract and elsewhere (e.g., "... had the highest risk ...").

PRESS

Sincerely, 

Richard Turner, PhD 

rturner@plos.org